# The Effects of Short- and Long-Term Spinal Brace Use with and without Exercise on Spine, Balance, and Gait in Adolescents with Idiopathic Scoliosis

**DOI:** 10.3390/medicina58081024

**Published:** 2022-07-29

**Authors:** Guilherme Erdmann da Silveira, Rodrigo Mantelatto Andrade, Gean Gustavo Guilhermino, Ariane Verttú Schmidt, Lucas Melo Neves, Ana Paula Ribeiro

**Affiliations:** 1Biomechanics and Musculoskeletal Rehabilitation Laboratory, Health Science Post-Graduate Department, Medicine School, University Santo Amaro, São Paulo 04829-300, SP, Brazil; gesilveira@prof.unisa.br (G.E.d.S.); ariane.schmidt@hotmail.com (A.V.S.); lmneves@prof.unisa.br (L.M.N.); 2Spine Group, Rehabilitation Clinic, São Paulo 13025-270, SP, Brazil; rodrigoandrade@usp.br (R.M.A.); gguilhermino99@gmail.com (G.G.G.); 3Department of Psychiatry, University of Sao Paulo, São Paulo 01246-903, SP, Brazil; 4Physical Therapy Department, School of Medicine, University of São Paulo, São Paulo 01246-903, SP, Brazil

**Keywords:** scoliosis, adolescents, brace, exercise, spine, gait, balance

## Abstract

*Background and Objectives*: Adolescent idiopathic scoliosis (AIS) is a prevalent spinal disorder in adolescents. Previous studies have shown biomechanical changes of the gait in the lower limb of AIS patients. To minimize the progression of scoliotic curvature, a spinal brace is used, which has been shown to be efficient. Usually, a brace is worn strictly for 20–22 h every day. To our knowledge, no study has assessed the short- and long-term effects of spinal brace use with or without an exercise program (6 months) to improve clinical and biomechanical parameters. The aim of our study was to verify the effects of short- and long-term spinal brace use, with or without an exercise program on the spine, body balance, and plantar load distribution during gait in AIS. *Materials and Methods*: A prospective randomized study was conducted with intention-to-treat analysis in forty-five adolescents diagnosed with AIS undergoing conservative treatment at a center specialized in spinal rehabilitation. Adolescents were evaluated at two stages of intervention: (1) spinal orthopedic brace, with acute use 24 h/day (*n* = 22) and (2) spinal orthopedic brace, with acute use between 15–18 h/day associated with a specific rehabilitation exercise protocol for six consecutive months (six months and 12 total sessions, *n* = 23). The evaluated parameters were: spine pain, using a visual analog scale (VAS); Cobb angle measurement using radiograph exams, as well as the Risser sign; and static balance and plantar pressure of the feet during gait, carried out using a pressure platform. *Results*: AIS patients showed significant improvements in the main scoliotic curvature, with a 12-degree reduction in Cobb angle pre- and post-short-term immediate use of spinal brace and a 5.3 degree correction after six months of spinal brace use in combination with specific exercises (long term). In addition, short- and long-term brace use with an exercise program showed a significant increase in anteroposterior and mediolateral balance and a reduction in plantar overload on the heel during gait, with an effect size between moderate and high. *Conclusions*: Intervention via the short- or long-term use of a spinal brace combined with specific exercises in adolescents with idiopathic scoliosis proved to be effective for correcting scoliotic curvature. In addition, intervention also showed improvements to the antero-posterior and mediolateral body balance and a reduction in the plantar load on the rearfoot region during gait, demonstrating effective mechanical action on the spine.

## 1. Introduction

The prevalence of adolescent idiopathic scoliosis (AIS) varies between 1.5 and 2%, higher in females with ages between 13 and 14 years [1,2]. AIS is defined as a three-dimensional deformity of the spine, with a Cobb angle equal to or greater than 10 degrees and changes to the curvatures in the sagittal plane [3]. Generally, the progression of curves is linked mainly to the rapid growth around puberty. Its progression is closely related to etiological factors such as sex, age of onset, degree of spine angulation, ventral overgrowth, and functional tethering of the spinal cord [1,2,4]. The progression of AIS can result in postural misalignment, reduced pelvic motion, imbalance in spine muscle activation, as well as balance and gait changes [5,6,7]. These changes can increase the progression of the disease and result in back pain and functional impairments with important social implications [5,8,9,10,11,12]. The balance reduction during gait is explained by the greater trunk rotation and asymmetry of spine forces [13,14]. Other studies show changes in the hip and knee motion, feet overload [10,15], increased gait speed [10,16], decreased cadence [17], and decreased step length [18]. These changes increase energy expenditure and reduce the aerobic conditioning of the patients [18,19], leading to neuromuscular and sensory disorders [20,21,22].

There are several conservative treatments aimed at AIS, specifically scoliosis-specific exercises (PSSE) using different exercise methods, such as the Schroth method, which uses the breath as the basis for its exercises, defining a system of body blocks; the Dobomed method, which utilizes the mobilization of the principal scoliotic curvature with the use of photographs and videos as feedback; the Barcelona scoliosis physiotherapy method based on the technique of Schroth’s exercises, with a similar body block system; individual functional scoliosis therapy (FITS), which uses body awareness as a basis for its exercises that are applied by patients in their homes (a multidisciplinary treatment, in association between physiotherapists, orthopedists, and psychologists), the scientific exercise approach to scoliosis (SEAS) method, active self-correction of the spine without external assistance, leading to postural correction through sensorineural stimuli; and exercise with the side-shift method [23,24,25].

Another essential conservative treatment to minimize the progression of scoliotic curvature is the use of spinal brace, which has been shown to be effective [26,27,28]. Indications for brace treatment are patients that are still growing, with a curve of 25° to 40° or with curves less than 25° and a documented progression of 5° to 10° in six months (progression of more than 1° per month). Patients with AIS of 20° to 25° with pronounced skeletal immaturity (Risser 0, Tanner 1 or 2) should also be treated immediately. Braces are generally be worn full-time until skeletal maturation and the end of bone growth [26]. There are several types and designs of braces aimed at the treatment of AIS; however, three-dimensional (3D)-modelled braces have been exhibiting more promising results in the literature, minimizing the progression of scoliotic curvature between 50 and 83, 8% [26,27,28] and avoiding surgical treatment when compared to two-dimensional (2D) orthoses or braces [28]. According to a recent narrative review in 2021, the outcomes of the Chêneau brace revealed success rates between 50% and 90% [29]. The Boston brace application is supported by success rates in the studies with 70% and 72% of patients [29,30,31]. This treatment has been one of the most widely used, not only to prevent the progression of the disease [32], but also to improve the quality of life of the affected patients [33,34].

The treatment time with the use of the brace varies from months to years, with average daily use of 20 h throughout the bone growth process, requiring the active participation of parents during the treatment period [29,32,33,34,35]. Indications for surgery exist, especially in patients with skeletal maturity [35,36]. Several studies have compared the effectiveness of the brace. Aulisa et al. (2012) [37] did not show progression of the scoliotic curve with the use of the Lyon and Milwaukee brace. Another study demonstrated that 77.8% of patients with AIS showed corrected scoliotic curvature [38]. Aulisa et al. (2015) [39], evaluating only the Lyon brace, found correction of scoliotic curvature in 85.5% of the patients. The use of the Lyon brace maintained the scoliotic curvature with no progression for over two years after the treatment [40]. With Cheneau’s brace in AIS patients, another study observed that after 1 year of wearing the brace, 25.3% of patients achieved an effective improvement and the progression of the Cobb angle was stabilized (below 50°) 22.8% of patients [41]. Other authors have demonstrated the effectiveness of the brace to stabilize scoliotic curvature in AIS [42,43].

Despite noting the benefits of the orthopedic brace for conservative treatment in greater scoliotic curvatures, there is little evidence of the isolated action of the brace in its first placement, which shows the clinical relevance of the present study in obtaining a greater understanding and emphasizing the clinical applicability of the brace in AIS. Thus, the purpose of the current study was to verify the effect of the spinal brace in both short- and long-term spinal brace use with or without an exercise program on the spine, body balance, and plantar load distribution during gait in AIS. The initial hypothesis was that short- and long-term spinal brace use combined with an exercise program would be effective for reducing the Cobb angle, increasing balance, and achieving a better distribution of plantar load during gait in AIS.

## 2. Materials and Methods

### 2.1. Design, Setting, Participants and Ethics

The current study is a prospective randomized study with intention-to-treat analysis, where all participants are randomized, included in the statistical analysis, and analyzed according to the group they were originally assigned regardless of what treatment they received. The adolescents were recruited from May to December 2020, from the Scientific Institute Specialized in Rehabilitation (REAB) of the city of Campinas and São Paulo (SP). Forty-five patients with AIS (clinical diagnosis performed by a doctor with specialty in the spine and confirmed by X-ray on a request for conservative treatment with brace and exercises), were recruited for intervention with immediate spinal brace (short term, *n* = 22) and spinal brace associated with a specific exercise protocol during a six-month period (long term, *n* = 23). The evaluations were carried out in the initial condition (T0, before starting treatment), after a 24-h immediate treatment with spinal brace (T1, called S4D), and after intervention with spinal brace associated with a specific exercise protocol (T6, six consecutive months totaling 12 sessions of treatment) (Figure 1).

The eligibility criteria for this study were: adolescent, between the ages of 10 and 17, a diagnosis of AIS confirmed by X-ray with a Cobb angle between 30 and 45° of the principal curvature (thoracic or lumbar, according to the Lenke classification), and a body mass index less than 35 kg/cm^2^. The exclusion criteria were: musculoskeletal disorders in the lower limbs related to the central and peripheral nervous system, diabetic neuropathies, rheumatoid arthritis, rigid foot deformities, previous or planned spinal surgery in the next twelve months, and mental disability. The adolescents also could not have prostheses and/or orthoses in the lower limbs or fractures in the previous 6 months and could not be receiving any other physical therapy treatment during the intervention period [43,44,45].

This study was previously submitted to the Research Ethics Committee of the University of Santo Amaro/SP, obtaining approval under number 2.729.155. Prior to participation, all adolescents or their respective legal guardians signed the free and informed consent form prepared in accordance with the Helsinki declaration and regulations.

### 2.2. Randomization and Blinding

First, the randomization schedule was prepared using Clinstat software by an independent researcher who was not aware of the numeric code for the intervention groups (brace or brace and exercise groups). A numeric block randomization sequence was kept in opaque envelopes. After the adolescents’ or their respective legal guardian’s agreement and assignment to participate in the research, allocation into groups was performed by another independent researcher, who was also unaware of the codes. Only the physiotherapist responsible for the clinical trial knew who was receiving the types of intervention. The initial and final clinical and biomechanical evaluations were carried out by an orthopedist who was blind to the patient’s allocation.

### 2.3. Evaluation of Clinical, Radiological, and Biomechanical Parameters

First, a questionnaire was applied to assess anthropometric and clinical characteristics. Soon after, an X-ray imaging examination of the spine was performed. To record the image, the adolescent remained in a static position with their feet flat on the floor and their forearms flexed with their wrists/hands positioned over the clavicles. Measurements were taken by a technician with experience in radiology. The radiographs were centered on T12 during inspiration, with a distance of 2 m between the film and the focus. All images were transferred to a computer as digital images and evaluated using Surgimap Spine imaging software (Nemaris Inc., New York, NY, USA) [45].

To calculate the thoracic and lumbar Cobb angle, as well as the main scoliotic curvature, the following procedure was used: first, the terminal vertebrae was identified and a line was drawn at the upper end of the cranial terminal vertebra and the other line drawn perpendicularly to the vertebral line. Next, a line was drawn through the lower end of the caudal vertebra of the curve, followed by a right angle to this line. The resulting Cobb angle was formed by the intersection of two lines perpendicular to the terminal vertebrae. All measurements were performed by the same orthopedist responsible for the study in [41]. Total thoracic kyphosis was measured using the T1 and T12 plateaus. Lumbar lordosis was measured using the angle formed between the upper endplate of L1 and S1, as described in a study carried out by Barsotti et al. (2021) [45]. In addition, evaluation of the Risser sign was estimated by a physician, according to Risser [45].

For the biomechanical assessment of static balance, a pressure platform (Loran^®^, Rome, Italy) was used, with the patient positioned in bipedal support with eyes open and arms alongside the body while remaining in static posture for 60 consecutive seconds. The variables measured were the anteroposterior and mediolateral postural sway (COP) as well as the velocity (m/s) and sway distance (cm) in relation to the center of gravity.

For the biomechanical evaluation of plantar pressure distribution during gait, the same pressure platform (Loran^®^ Sensor Medica Inc., Rome, Italy) was used, with dimensions of 3240 mm in length, 620 mm in width, 20 mm in height, and 29 kg in weight. The equipment includes homogeneously distributed resistive pressure sensors (4 sensors/cm^2^). The platform was connected to a desktop notebook to transmit data that were collected at a frequency of 100 Hz. The adolescents performed the walk at a pre-established cadence. To ensure that they had reached this cadence, plantar pressure acquisitions were monitored using a stopwatch. The adolescents were familiarized with the collection environment and instruments to reduce the retroactive effect. The standardized instructions were given by the same examiner (physiotherapist spine specialist) for all timepoints of treatment. After familiarization, the teenagers walked on a flat synthetic rubber track for a distance of 20 m. The steps comprised in the intermediate 10 m were timed and validated for the analysis, totaling approximately 12 steps captured in six rounds of the track to record the foot support on the platform. The plantar pressure variables analyzed and measured were: (1) maximum peak pressure value per selected area—the maximum pressure value (expressed in kPa); (2) maximum force—the value of maximum force (expressed in N); and (3) contact area—the area in which the sensors were activated (pressed) in each step (expressed in cm^2^). All plantar pressure variables were analyzed in 4 plantar areas of the feet. For this purpose, the foot was divided into four areas: medial and lateral hindfoot (30% of foot length), midfoot (30% of foot length), and forefoot and toes (40% of foot length) [46].

### 2.4. Treatment with Spinal Brace Immediate (Short-Time 24 h Isolated)

The International Society on Scoliosis Orthopaedic and Rehabilitation Treatment (SOSORT) establishes that conservative treatment with a brace and specific exercises is beneficial for the progressive correction and prevention of scoliotic curvature [40,47]. The spinal brace (S4D) was used in short-term (immediate effect during 24 h) and long-term use of the spinal brace (3D) associated with specific exercises as part of a conservative treatment of AIS [32,40,41,44,47,48]. Prior to conservative treatment with a brace, the patient previously received a clinical diagnosis from their physician confirmed by radiography, with an average time of 3 months for the completion and elaboration of the brace (Figure 2).

The three-dimensional spinal brace used was the S4D brace, produced in Brazil. The S4D brace was designed and manufactured using the computer-aided CAD/CAM system and designed using the Rodin4D computational system, which is based on the correction of deformities of compression forces at three points to improve the treatment results and prevent the appearance of secondary curves. The CAD/CAM measurements for the fabrication of the S4D brace were made using the latest generation of digital scanners. This system makes it possible to obtain a personalized model for each patient with precise virtual correction for subsequent fabrication. The process involved professionals from different areas, including doctors, physiotherapists, and designers.

### 2.5. Specific Exercise Program Protocol

The intervention program with specific exercises lasted for six consecutive months, and consisted of two monthly sessions with a duration of 40 min for each session, combined with the use of the spinal brace (average use of 18–20 h daily). The intervention protocol was divided into three phases, according to the improvement of the participant, as follows: (1) training with axial growth exercises and spinal self-correction in the frontal and sagittal planes; (2) training of rotation, stabilization, mobilization, and stretching exercises; and (3) motor coordination training: dual task, functionality, and balance. It is worth mentioning that all the exercises proposed in the intervention program were based on evidence from the literature in systematic review studies and clinical trials [32,40,41,48]. The complete description of the intervention protocol, as well as the method of performing the exercises, the progression criteria in each phase, and the volume and duration of the exercises for each intervention phase are presented in Table 1, Table 2 and Table 3. After completion of the intervention program, patients were re-evaluated.

### 2.6. Statistical Analysis

The sample size calculation on 55 patients was based on the mean Cobb angle of principal curvature using the G-Power 3.0 software and considered a moderate effect size (F = 0.25), a power of 80%, and a significance level of 5%. Normality of the data was tested using the Shapiro–Wilks test and parametric tests were applied after confirmation. Anthropometric characteristics, clinical, radiological, and biomechanical variables (TO—baseline, T1—immediate brace and T6—brace associated with specific exercises) were verified using one-way repeated measures analysis of variance (ANOVA) followed by Tukey’s post-hoc test. To calculate the effect size, Cohen’s d test was used, in which the values of 0.2, 0.5, and 0.8 were considered small, medium, and large effect sizes, respectively. For all analyses, a significance level of 5% was adopted.

## 3. Results

Initially, 55 adolescents with AIS volunteered to participate in this study. Eight were excluded by the eligibility criteria and two dropped out of treatment due to the distance from their residence to the rehabilitation clinic. A total of 45 adolescents with AIS participated and completed the treatment before and after spinal brace use with or without an exercise program (Figure 1). The anthropometric and skeletal maturity characteristics of patients with AIS do not show significant differences between the different treatment timepoints (Table 4).

Comparing the radiographic parameters of the scoliotic curvature, a reduction of 12.0 degrees in the Cobb angle with a high and significant effect size can be observed after immediate use of the brace (short term) and a reduction of 5.3 degrees can be observed after six months of brace use with specific exercises (long term). In relation to the thoracic Cobb angle, a correction of 6.3 degrees was obtained after immediate use of the brace and of a correction of 4.5 degrees was obtained after six months of brace use with specific exercises. In the lumbar Cobb angle, the correction was 9.5 degrees after immediate use of the brace and 7.2 degrees after six months of using the brace with specific exercises, as can be observed in Table 5.

Table 6 shows the biomechanical aspects of the distribution of plantar load on the feet during the gait of adolescents with AIS, where it was found that the contact area was reduced in the forefoot. In addition, the pressure peak was reduced in all areas of the feet (forefoot, midfoot and medial and lateral hindfoot), and the maximum strength in the forefoot and hindfoot was reduced after the immediate use of the brace and after six months of the brace with the specific exercises (Table 6). Effect sizes were found to be from high to moderate in reducing the peak pressure in the midfoot (d = 1.30; d = 1.18) and maximum force in the forefoot (d = 0.58; d = 0.34) and medial rearfoot (d = 0.34; d = 0.31) after both timepoints of intervention with spinal brace (short and long term) (Table 6).

During static posture, it was observed that the contact area of the medial and lateral rearfoot was increased after the immediate use of the brace and after six months of treatment with the brace and specific exercises. On the other hand, the peak pressure and maximum force on the medial and lateral hindfoot region were reduced with a small to moderate effect size after both treatment timepoints. Maximum force was also reduced in the midfoot region for both timepoints of spinal brace treatment (Table 7).

For the last parameter, body balance, a significant increase with a moderate to high effect size of body sway in relation to the center of gravity as well as body sway of the feet (right and left), only after six months of using the brace with specific exercises. 

On the other hand, the anteroposterior and medio-lateral oscillations, as well as the circumference area of oscillation, showed to be significantly increased with the immediate use of the brace and after six months of the brace with specific exercises. The oscillation speed was not different between the different timepoints of AIS treatment (Table 8).

## 4. Discussion

The purpose of this study was to verify the effect of short- and long-term spinal brace (24 h) use with and without an exercise program on posture, body balance, and plantar load distribution during gait in AIS. The main results showed that the Cobb angle was reduced by an average of 12°, with a high and significant effect size after immediate use of the brace (short-term) and an average of 5.3° after six months of use of the brace associated with specific exercises (long term). As for the radiographic parameters of the thoracic Cobb angle and lumbar Cobb angle, the correction was greater after immediate use of the brace (6.3° degrees and 9.5° degrees, respectively) and lower after six months of using the brace with the specific exercises (4.5° and 7.2°, respectively).

During gait, a reduction was observed in the contact area of the forefoot, in the peak pressure and maximum force on the forefoot and rearfoot (medial and lateral), as well as in the peak pressure on the midfoot for both treatment timepoints with immediate and long-term use of the brace (combined with specific exercises). In the static posture, it was observed that the contact area of the rearfoot (medial and lateral) increased, while the peak pressure and maximum force reduced after the use of the immediate brace and in long-term use, after six months of the use of the brace with the specific exercises, as well as the maximum force on the midfoot region, showing a better distribution of the plantar load. In body balance, a significant increase in body sway can be observed in relation to the center of gravity, as well as anteroposterior and medio-lateral sway and sway distance for both timepoints of brace use with and without an exercise specific program in EIA.

In this study, the immediate use of the spinal brace (24 h) was more effective in correcting the Cobb angle than the long-term use, after six months of wearing the spinal brace associated with specific exercises. According to the literature, studies that aimed to evaluate the Cobb angle before and after use of the spinal brace (18 to 23 h a day) during a period of 2 to 3 years observed an average correction of 3–6 degrees [39,40,41,47,48,49]. A systematic review revealed that the use of the brace with exercises increases the effectiveness of conservative treatment when compared to exercises program alone [34].

The difference between this study and previous research is the comparison of the immediate effect of the brace and its long-term use (six months) with specific exercises to correct the Cobb angle in AIS. Another point that differentiates this study is that we were careful to make the spine brace according to the classification and specificity of the scoliotic curvature, which is three-dimensional. There are several types of spine braces intended for the treatment of AIS; however, currently three-dimensional (3D) brace models have been highlighted in the literature for their promising results in minimizing the worsening of the scoliotic curvature [50,51,52,53].

According to the evidence in the literature, the effectiveness of the spinal brace in AIS depends on the Cobb angle correction index, the time of use, patient compliance, the magnitude of the Cobb angle, and the stage of growth of the adolescent [50,54]. In this study, we standardized some of these points, such as the immediate effect of the brace correction, the time of use (between 12–18 h daily), and skeletal maturity by the risser sign, factors that favored maintaining the effectiveness of the brace correction over a long-term period combined with specific exercises. According to Clin et al. (2010) [55], when evaluating three patients with idiopathic scoliosis in 3D computer models, the authors concluded that the brace allowed greater correction of the Cobb angle in the flexible spine simulation model. In this study, we did not evaluate the Cobb angle and the characteristics of the spine in rigid or flexible models, but the results showed a positive effect in the correction of the Cobb angle with the immediate use of the spinal brace (mean 12.0 degrees) and in the long term (after six months), with a corrective effect of 44.4%.

Another important issue discussed in the literature and in review studies [13,14] is the lack of understanding of the biomechanical aspects of gait and balance in AIS, after the effect of treatment with the short and long-term use of the brace, since many observational studies showed an increase in the plantar load on the forefoot and its association with body imbalance in adolescents with the disease [56,57]. According to Catan et al. (2020) [57], the increase in the plantar load of the feet may favor an increase in the Cobb angle in AIS. Only one study evaluated the gait of patients with scoliosis using a stabilizing orthosis, and the results showed greater spinal stability during walking [58]. The difference between this study and previous research is the analysis of the biomechanical aspects of plantar load distribution after the use of the spinal brace, in the short- and long-term use (with specific exercises) for AIS. The results showed a reduction in the plantar load on the forefoot, midfoot, and rearfoot (medial and lateral), favoring better balance control, and a reduction in the forces imposed on the spine, which can help minimize the progression of the Cobb angle.

The efficacy of conservative approaches to scoliosis treatment is still an open debate. Alternative forms of non-surgical treatment and braces, such as chiropractic or osteopathy, acupuncture, specific exercises, or other manipulative treatments, are still being verified through clinical trials in their ability to reduce the Cobb angle and improve the motor and functional activities of patients with AIS. Although the subject of debate, most experts agree that physiotherapy with a specific exercise program in conjunction with brace treatment is beneficial. The triad of out-patient physiotherapy, intensive in-patient rehabilitation, and bracing has proven effective in conservative scoliosis treatment in central Europe [29,44,59,60]. However, there are still no studies in the literature with clinical trials of patients with AIS in Brazil, which—in addition to verifying the correction of the Cobb angle of the main scoliotic curvature—verify the responses of the treatment for gait changes with the best adjustment of the vectors of vertical force received by the lower limbs and spine, as well as the improvement of body balance. In this study, we can observe a reduction in the plantar load on feet, indirectly favoring a reduction in the forces imposed on the spine, which can help minimize the progression of the Cobb angle.

In relation the plantar load in static posture, an increase in the contact area on the rearfoot (medial and lateral) was observed, which favored a better distribution of the plantar load to the midfoot and forefoot regions and thus, a better performance of the body balance. This can be explained by the increase in body sway in relation to the center of gravity, as well as in the anteroposterior and medio-lateral sways and the area of sway of the patients after treatment with the spinal brace (short- and long-term use). According to the literature, imbalance worsens body instability on the side of the main scoliotic curvature, favoring the progression of AIS [61,62].

A limitation of this study is the non-differentiation of the different types of scoliotic curvature and their interaction with immediate and long-term brace use associated with program exercises. Future studies that understand this issue can help and favor improved planning of the rehabilitation process for balance and gait of patients with AIS.

## 5. Conclusions

The intervention with the use of the spinal brace, in short-term usage and long-term usage combined with specific exercises in adolescents with idiopathic scoliosis proved to be effective in correcting the scoliotic curvature (Cobb angle). In addition, the intervention improved the antero-posterior and mediolateral body balance and reduced the plantar load on the rearfoot region during gait, demonstrating effective mechanical action on the spine.

## Figures and Tables

**Figure 1 medicina-58-01024-f001:**
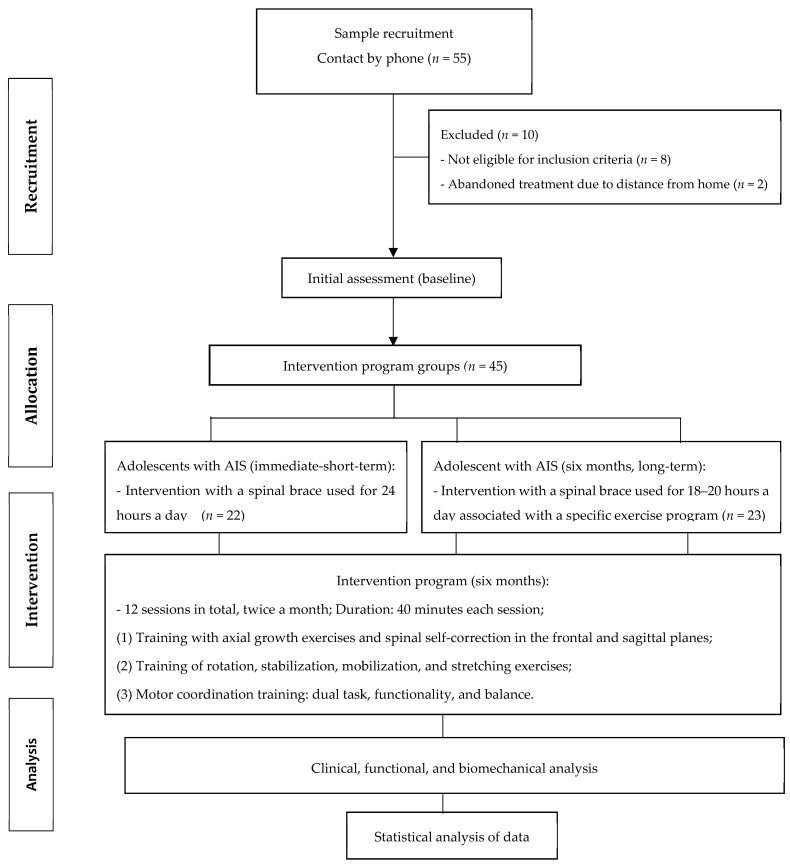
Intervention Protocol in adolescents with AIS.

**Figure 2 medicina-58-01024-f002:**
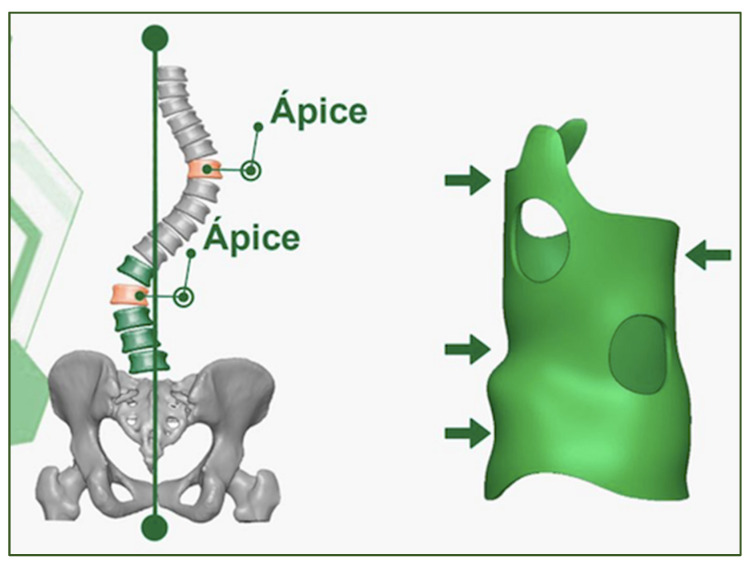
The spinal brace (S4D) used during short- and long-term use as an intervention program for the treatment of AIS, the arrows demonstrate the support and correction points of the brace on the spine.

**Table 1 medicina-58-01024-t001:** Intervention protocol with specific exercises for axial growth and spinal self-correction (frontal and sagittal planes) for adolescents with idiopathic scoliosis (AIS).

Training	Description of the Exercises	Execution of the Specific Exercises
Training with axial growth exercises and spinal self-correction (frontal and sagittal planes)	Axial growth with breathing Spinal self-correction	Sitting, with feet and knees apart and aligned forward with hands pushing the legs for axial trunk growth with scoliotic curvature correction associated with inspiration. Sitting, with feet and knees apart and aligned forward, with one hand on the back of the head and the other contralateral to the thoracic curve, pushing the leg for axial growth of the trunk associated with respiratory expansion to correct the convexity of the curvature. Sitting, with feet and knees apart and aligned forward, with symmetrical upper limbs with a stick followed by a resistive elastic band associated with breathing to correct the scoliotic curvature.
Intensity Parameters	Frequency	2 sessions/monthly
	Repetition	10 maintaining three respiratory cycles
	Duration Rest	3 to 5 min 30 s every 5 reps
Progression Parameters	Symptoms	No pain or muscle fatigue

**Table 2 medicina-58-01024-t002:** Intervention protocol with specific rotation, stabilization, mobilization, and stretching exercises for adolescents with idiopathic scoliosis (AIS).

Training	Description of the Exercises	Execution of the Specific Exercises
Training of rotation, stabilization, mobilization, and stretching exercises	Rotation with stabilization Rotation with decompression Rotation with mobilization and strength resistance	Sitting, with feet and knees apart and aligned forward, hands on the stick (vertical, diagonal, and horizontal direction) supported on the concave side of the scoliotic curve, and keeping the elbows perpendicular to the stick, perform correction of the curvature in the sagittal plane associated with breathing of chest expansion with trunk rotation to the concave side of the curvature. Sitting, with feet and knees apart and aligned forward, hand up with shoulder abducted 180 degrees from the concave side of the thoracic curve to decompress, and the other hand pushing the thigh for axial growth of the body, perform sagittal plane rotation correction associated with chest expansion breathing with trunk rotation to the concave side of the curvature. Sitting, with feet and knees apart and aligned forward, hand on the vertical stick on the concave side of the scoliotic curvature and the other hand holding the open door handle, and keeping the elbow perpendicular to the floor, perform rotation correction in the sagittal plane associated with chest expansion breathing with trunk rotation to the concave side of the curvature. Standing, with the feet facing forward, the lower limb on the concave side of the scholtic curvature in front in hip flexion, and the foot holding the end of the elastic band, and hands holding the other end of the elastic band with the elbows high, perform rotation correction in the sagittal plane associated with chest expansion breathing with trunk rotation to the concave side of the curvature.
Intensity Parameters	Frequency	2 sessions/monthly
	Repetition	10 maintaining three respiratory cycles
	Duration Rest	3 to 5 min 30 s every 5 reps
Progression Parameters	Symptoms	No Pain or Muscle Fatigue

**Table 3 medicina-58-01024-t003:** Intervention protocol with specific motor coordination exercises: dual task, functionality, and balance for adolescents with idiopathic scoliosis (AIS).

Training	Description of the Exercises	Execution of the Exercises
Motor coordination training: dual task, functionality and balance	Coordination with lower limb forward and feet on the floor in static posture Coordination with the front and back lower limb simulating dynamic gait with or without footrests on proprioceptive disc	Standing, with hands and forearms resting on the door frame and shoulders in 110° of abduction, the lower limb on the convex side of the scoliotic curve forward with the knee in flexion, and the lower limb on the concave side of the curve backward with the knee in extension, perform rotation correction with maintenance of curvature in the sagittal plane associated with breathing (thoracic expansion). Standing, with hands resting on the waist and open elbows, feet parallel and aligned forward, and lower limb forward on the convex side of the scoliotic curvature, perform rotation correction and maintenance of curvature in the sagittal plane. Next, perform a step forward of the lower limb on the convex side of the curvature, with the knee in flexion and feet on the floor progressing to the proprioceptive disc, and the contralateral (concave side of the curve) backwards, with the knee in extension. Then, perform rotation correction with maintenance of curvature in the sagittal plane associated with breathing (thoracic expansion).
Intensity Parameters	Frequency	2 sessions/monthly
	Repetition	10 maintaining three respiratory cycles
	Duration Rest	3 to 5 min 30 s every 5 reps
Progression Parameters	Symptoms	Completed repetitions with cyclic respiratory parameters

**Table 4 medicina-58-01024-t004:** Anthropometric and skeletal maturity profile of adolescents with idiopathic scoliosis between different timepoints: T0—baseline, T1—after immediate use of the spinal orthopedic brace (short term, 24 h), and T6—after six months of brace use associated with specific exercises (long term).

Characteristics Anthropometric	T0 Baseline	T1 Immediate Brace (Short Term)	T6 Brace and Exercise (Long Term)	*p*
Age (years)	13.2 ± 1.6	13.4 ± 1.6	13.6 ± 1.7	0.687
Stature (m)	1.5 ± 0.1	1.5 ± 0.2	1.5 ± 0.2	0.339
Body mass (kg)	49.2 ± 8.0	49.1 ± 8.0	50.2 ± 8.8	0.858
Body Mass Index BMI (kg/cm^2^)	16.5 ± 4.5	16.4 ± 3.5	15.8 ± 2.6	0.840
Risser (sign)	2.0 ± 1.7	2.0 ± 1.7	2.0 ± 1.4	0.735

One-way repeated measures ANOVA test between the different timepoints T0, T1, and T6, considering significant differences to be *p* < 0.05.

**Table 5 medicina-58-01024-t005:** Comparisons of radiographic parameters between the different timepoints: T0—baseline, T1—after immediate use of the spinal orthopedic brace (short term, 24 h) and T6 after six months using brace associated with specific exercises (long term) in adolescents with idiopathic scoliosis.

Radiographic Parameters	T0 Baseline	T1 Brace (Short Term)	T6 Brace with Exercise (Long Term)	d ^(1–2)^	d ^(1–3)^	*p*
Cobb angle (main curvature, degrees)	40.3 ± 5.0	28.3 ± 7.1	35.1 ± 8.8	1.30	0.62	<0.001 *^,#^
Thoracic Kyphosis Angle (degrees)	34.2 ± 13.9	27.9 ± 10.8	29.7 ± 13.5	0.50	0.98	0.042 *^,#^
Lumbar lordosis angle (degrees)	32.1 ± 13.5	22.6 ± 7.0	24.9 ± 14.6	0.88	0.51	0.025 *^,#^

Values are calculated using one-way repeated measures ANOVA tests between the different timepoints T0, T1, and T6 with Tukey’s post-hoc test, considering significant differences to be *p* < 0.05. *, significant difference between T0 and T1; ^#^, significant difference between T0 and T6. Cohen’s d test was used to check effect size.

**Table 6 medicina-58-01024-t006:** Comparisons of plantar pressure distribution during gait between the different timepoints: T0—baseline, T1—after immediate use of the spinal orthopedic brace (short term, 24 h), and T6—after six months using a brace associated with specific exercises (long term) in adolescents with idiopathic scoliosis.

Plantar Pressure during Gait	Regions of the Feet	T0 Baseline	T1 Brace (Short Term)	T6 Brace with Exercise (Long Term)	d ^(1–2)^	d ^(1–3)^	*p*
Contact Area (cm^2^)	Forefoot	11.6 ± 5.0	9.7 ± 3.2	8.1 ± 1.4	0.10	0.10	0.003 *^,#^
	Midfoot	9.5 ± 6.7	11.2 ± 5.4	8.8 ± 5.6	0.23	0.20	0.393
	Medial Rearfoot	16.5 ± 3.6	15.4 ± 3.2	15.5 ± 3.1	0.10	0.10	0.316
	Lateral Rearfoot	16.8 ± 3.4	15.6 ± 3.2	15.7 ± 3.3	0.29	0.30	0.296
Peak Pressure (KPa)	Forefoot	245.6 ± 41.7	237.8 ± 45.2	237.3 ± 46.5	0.17	0.18	0.001 *^,#^
	Midfoot	82.8 ± 5.3	74.8 ± 6.5	77.3 ± 4.5	1.30	1.18	0.025 *^,#^
	Medial Rearfoot	280.5 ± 58.5	265.6 ± 65.8	271.6 ± 48.1	0.23	0.16	0.001 *^,#^
	Lateral Rearfoot	268.5 ± 53.3	255.8 ± 65.8	256.8 ± 45.0	0.21	0.23	0.001 *^,#^
Maximum force (N/kg)	Forefoot	10.8 ± 3.0	9.2 ± 2.4	9.8 ± 2.8	0.58	0.34	0.045 *^,#^
	Midfoot	4.8 ± 2.8	4.7 ± 2.6	4.6 ± 2.9	0.11	0.10	0.470
	Medial Rearfoot	23.6 ± 9.7	20.8 ± 6.3	20.9 ± 7.1	0.34	0.31	0.045 *^,#^
	Lateral Rearfoot	21.1 ± 7.9	19.3 ± 5.8	18.8 ± 7.5	0.26	0.30	0.038 *^,#^

Values are calculated using one-way repeated measures ANOVA tests between the different timepoints T0, T1, and T6 with Tukey’s post-hoc test, considering significant differences to be *p* < 0.05. *, significant difference between T0 and T1; ^#^, significant difference between T0 and T6. Cohen’s d test was used to check effect size.

**Table 7 medicina-58-01024-t007:** Comparisons of static posture between the different timepoints: T0—baseline, T1—after immediate use of the spinal orthopedic brace (short term, 24 h) and T6 after six months using the brace associated with exercises (long term) specific in adolescents with idiopathic scoliosis.

Static Posture	Regions of the Feet	T0 Baseline	T1 Immediate Brace (Short Term)	T6 Brace and Exercise (Long Term)	d ^(1–2)^	d ^(1–3)^	*p*
Contact Area (cm^2^)	Forefoot	7.4 ± 2.7	6.6 ± 2.6	6.2 ± 2.6	0.30	0.45	0.143
	Midfoot	7.0 ± 2.5	7.1 ± 2.6	6.4 ± 2.7	0.14	0.23	0.116
	Medial Rearfoot	15.4 ± 2.8	16.9 ± 3.6	16.7 ± 2.7	0.46	0.47	0.035 *
	Lateral Rearfoot	15.7 ± 2.4	16.6 ± 3.2	16.1 ± 2.6	0.31	0.16	0.047 *
Peak Pressure (KPa)	Forefoot	77.8 ± 47.5	66.3 ± 43.0	55.5 ± 40.2	0.25	0.50	0.134
	Midfoot	37.9 ± 19.1	35.6 ± 17.6	30.4 ± 12.3	0.12	0.46	0.133
	Medial Rearfoot	174.4 ± 84.9	205.8 ± 79.6	168.3 ± 78.9	0.38	0.17	0.011 *
	Lateral Rearfoot	159.2 ± 76.3	185.7 ± 74.7	148.0 ± 74.1	0.35	0.11	0.016 *
Maximum force (N/kg)	Forefoot	2.8 ± 0.8	2.1 ± 0.9	3.3 ± 1.9	0.82	0.34	0.106
	Midfoot	2.0 ± 0.6	1.4 ± 0.6	1.5 ± 0.8	1.0	0.70	0.008 *^,#^
	Medial Rearfoot	11.6 ± 6.3	13.2 ± 5.8	10.4 ± 5.0	0.42	0.22	0.035 *^,#^
	Lateral Rearfoot	8.8 ± 3.3	9.8 ± 4.1	7.5 ± 3.9	0.53	0.20	0.001 *^,#^

Values are calculated using one-way repeated measures ANOVA tests between the different timepoints T0, T1, and T6 with Tukey’s post-hoc test, considering significant differences to be *p* < 0.05. *, significant difference between T0 and T1; ^#^, significant difference between T0 and T6. Cohen’s d test was used to check effect size.

**Table 8 medicina-58-01024-t008:** Comparisons of body balance between the different moments: T0—baseline, T1—after immediate use of the spinal orthopedic brace (short term, 24 h) and T6 after six months using the brace associated with specific exercises (long term) in adolescents with idiopathic scoliosis.

Body Balance Parameters	T0 Baseline	T1 Immediate Brace (Short Term)	T6 Brace and Exercise (Long Term)	d ^(1–2)^	d ^(1–3)^	*p*
Body sway to the center of gravity	230.4 ± 48.8	334.0 ± 31.9	351.8 ± 29.5	2.5	3.0	0.018 ^#^
Right foot body sway	114.1 ± 35.2	270.4 ± 39.7	316.6 ± 43.8	2.4	5.0	0.048 ^#^
Left foot body sway	118.1 ± 48.0	137.3 ± 26.8	175.7 ± 32.5	0.50	1.4	0.037 ^#^
Anteroposterior sway	0.73 ± 0.8	0.76 ± 0.2	1.58 ± 0.9	0.50	0.99	0.002 *^,#^
Mediolateral sway	2.50 ± 0.2	3.78 ± 0.4	4.0 ± 0.9	3.7	2.3	0.007 *^,#^
Distance (cm)	389.0 ± 30.4	461.1 ± 33.8	505.4 ± 31.9	2.2	3.7	0.038 *^,#^
Speed (m/sec.)	0.010 ± 0.1	0.020 ± 0.5	0.021 ± 0.5	0.02	0.03	0.215

Values are calculated using one-way repeated measures ANOVA tests between the different timepoints T0, T1, and T6 with Tukey’s post-hoc test, considering significant differences to be *p* < 0.05. *, significant difference between T0 and T1; ^#^, significant difference between T0 and T6. Cohen’s d test was used to check the effect size.

## Data Availability

The datasets generated and/or analyzed during the current study are not publicly available due to limitations of ethical approval involving the patient data and anonymity but are available from the corresponding author (apribeiro@alumni.usp.br) on reasonable request.

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
