# Peer review of "The Effects of Short- and Long-Term Spinal Brace Use with and without Exercise on Spine, Balance, and Gait in Adolescents with Idiopathic Scoliosis"

_medicina, 2022, doi:10.3390/medicina58081024_

Round 1

Reviewer 1 Report

I think it is an excellent work, in line with previous literature. And I think it is a work that benefits patients with AIS. I do not consider it necessary to make any major corrections, although I would like to clarify something:

- What exactly is the study design? 

- Is it an interventional study?

- What is the randomization method?

- Is there any blinding?

Author Response

São Paulo, 14th of July 2022

Dear Chief Editor

Prof. Dr. Edgaras Stankevičius

Editor and Editorial Board Member of Medicina

We, the authors, would like to resubmit the paper The effect of short and long-term spinal brace with and without exercise on spine, balance and gait in adolescent with idiopathic scoliosis” (1761918-1) in a revised form, as suggested. We are also sending a covering letter responding to the reviewer’s comments on a point-by-point basis. Bellow, we followed your comments and answered each one also in a point-by-point basis. The authors would like to thank the editor for the careful revision and constructive comments/suggestions on our manuscript that certainly contributed for a better version of it.

Yours sincerely,

The authors

Responses to the Reviewer #1:

Submission: MEDICINA (ID 1761918-1)

Reviewer #1:

We thank the reviewer for the constructive comments on our manuscript. Your suggestions and remarks have helped us to reflect on the manuscript and make a better version. We appreciate your suggestions and carefully considered every one of your comments and we made the appropriate changes. Below, we responded to your remarks on a point-by-point basis and inserted the corrections to the new version of the manuscript (underlined parts). Some of your comments will be discussed here. First your comment is given in bold; subsequently we provide our answer.

  1. Comments and Suggestions for Authors: I think it is an excellent work, in line with previous literature. And I think it is a work that benefits patients with AIS.

Answer: We appreciate your comment and all your attention and care in reviewing this manuscript. We are very honored by your review.

  1. I do not consider it necessary to make any major corrections, although I would like to clarify something: - What exactly is the study design? Is it an interventional study?

Answer: We agree with reviewer and this study was a prospective randomized study with intention-to-treat analysis, where all participants who are randomized are included in the statistical analysis and analyzed according to the group they were originally assigned, regardless of what treatment they received. For better understanding, we have added this information in the abstract and methodology with the citation of the reference below, which better details this type of study. Page 1 (line 19); Page 3 (line 117-121); Page 13 (line 522-523). Underlined parts. 

  • McCoy CE. Understanding the Intention-to-treat Principle in Randomized Controlled Trials. West J Emerg Med. 2017 Oct;18(6):1075-1078. doi: 10.5811/westjem.2017.8.35985.

  1. What is the randomization method? Is there any blinding?

Answer: We thank you very much for your precious time and dedication to reviewing our manuscript. We agreed with the reviewers and added more details and a topic about randomization and blinding in the text. Page 5 (line 148-157). Underlined parts. 

Reviewer 2 Report

There are many grammatical and typographical errors throughout the manuscript. Please rectify 

Abstract: The background does not read well. Please modify The sample size is very small What was the baseline comparison between the groups?   Intro  Calling adolescent idiopathic scoliosis as an equivalent of juvenile and early onset scoliosis is a factual blunder. These are totally different entities. Wrong message altogether Growth period is not towards the end of juvenile period Very long and not focussed. Pls modify Instead of mentioning the indications of bracing, the authors have out forth indications for all treatment options. Pls modify the sentence   Methods Which brace was used? Not clear in the methods section The exact protocol followed needs to be better elaborated. It is at the end of the methods section. This is in fact the fundamental issue discussed in the study. Too many parameters are studied, which makes the reading hard for the readers. Please reorganise the methods section The methodology is very simplistic and therefore includes a very heterogeneous population of scoliosis patients.     Results The authors have tried to include too many parameters for the analysis. If foot pressure point analysis was the primary parameter to be assessed, they should focus more on presenting the data more clearly. In the current format, the details are hard to follow   There is no control group for comparison   Discussion Why did the authors choose a combination of exercises and bracing as the protocol. What is the literature evidence that a combination of these approaches is better than only one of them?     The discussion needs to be more focused on the conservative approaches to AIS

Author Response

São Paulo, 14th of July 2022

Dear Chief Editor

Prof. Dr. Edgaras Stankevičius

Editor and Editorial Board Member of Medicina

We, the authors, would like to resubmit the paper The effect of short and long-term spinal brace with and without exercise on spine, balance and gait in adolescent with idiopathic scoliosis” (1761918-1) in a revised form, as suggested. We are also sending a covering letter responding to the reviewer’s comments on a point-by-point basis. Bellow, we followed your comments and answered each one also in a point-by-point basis. The authors would like to thank the editor for the careful revision and constructive comments/suggestions on our manuscript that certainly contributed for a better version of it.

Yours sincerely,

The authors

Responses to the Reviewer #2:

Submission: MEDICINA (ID 1761918-1)

Reviewer #2:

We thank the reviewer for the constructive comments on our manuscript. Your suggestions and remarks have helped us to reflect on the manuscript and make a better version. We appreciate your suggestions and carefully considered every one of your comments and we made the appropriate changes. Below, we responded to your remarks on a point-by-point basis and inserted the corrections to the new version of the manuscript (underlined parts). Some of your comments will be discussed here. First your comment is given in bold; subsequently we provide our answer.

  1. There are many grammatical and typographical errors throughout the manuscript. Please rectify.

Answer: We appreciated your comment and strongly apologize for the grammatical and typographical errors in the manuscript. We have submitted the manuscript to a professional service for reviewing English as a second language (Scribendi). We hope it achieves the standards of this respectful Journal.

  1. Abstract: The background does not read well.

Answer: We thank the reviewer and we agree that this description needs a better explanation. Page 1 (line 14-19). Underlined parts.

  1. Please modify the sample size is very small. What was the baseline comparison between the groups?

Answer: We appreciate your feedback and comment. We apologize, but the sample number was wrongly described in the calculation of the statistical analysis. We corrected this and added the numbers of patients in each group and also explained the baseline and monitoring times for the groups. Page 2 (line 25-26); Page 3 (line 130-133); Page 7 (line 277). Underlined parts. 

  1. Introduction: Calling adolescent idiopathic scoliosis as an equivalent of juvenile and early onset scoliosis is a factual blunder. These are totally different entities. Wrong message altogether Growth period is not towards the end of juvenile period. Very long and not focussed. Pls modify Instead of mentioning the indications of bracing, the authors have out forth indications for all treatment options. Pls modify the sentence?

Answer: We agree with the reviewer and apologize for this error. We rewrote the sentences for better understanding and also added spinal brace indications. In addition, the reference below has been added. Page 2 (line 50-53); Page 2 (line 84-89). Underlined parts. 

“Kaelin AJ. Adolescent idiopathic scoliosis: indications for bracing and conservative treatments. Ann Transl Med. 2020 Jan;8(2):28. doi: 10.21037/atm.2019.09.69.”

  1. Methods: Which brace was used? Not clear in the methods section. The exact protocol followed needs to be better elaborated. It is at the end of the methods section. This is in fact the fundamental issue discussed in the study. Too many parameters are studied, which makes the reading hard for the readers. Please reorganise the methods section. The methodology is very simplistic and therefore includes a very heterogeneous population of scoliosis patients.

Answer: We thank and agree with the reviewer. We have added details of spinal brace and spinal brace exercise protocol on figures 3, 4 and 5 for better understanding of exercise protocol for AIS patients. Page 6 (line 251-259); Page 7 (figure 3); Page 8 (figure 4) and Page 9 (figure 5). Underlined parts. 

  1. Results: The authors have tried to include too many parameters for the analysis. If foot pressure point analysis was the primary parameter to be assessed, they should focus more on presenting the data more clearly. In the current format, the details are hard to follow. There is no control group for comparison.

Answer: The current study was a prospective randomized study with intention-to-treat analysis, where all participants who are randomized are included in the statistical analysis and analyzed according to the group they were originally assigned, regardless of what treatment they received. Thus, the research design was the comparison of outcome before and after a planned two intervention protocol without the use of a control group (known as the pre/post design), but we also aimed to compare the groups of interventions performed for patients with AIS. All specific statistical analyzes are detailed in the legends of the results tables (1, 2, 3, 4 and 5). We hope to have answered the reviewer's questions with explanations. Page 10 (table 1 line 326-331); Page 11 (table 2 342-348 and table 3 358-364); Page 12 (table 4 371-376 and table 5 line 385-391). Underlined parts. 

  1. Methods: Which brace was used? Not clear in the methods section. The exact protocol followed needs to be better elaborated. It is at the end of the methods section. This is in fact the fundamental issue discussed in the study. Too many parameters are studied, which makes the reading hard for the readers. Please reorganise the methods section. The methodology is very simplistic and therefore includes a very heterogeneous population of scoliosis patients.

Answer: We appreciate your comment and all your attention and care in reviewing this manuscript. We are very honored by your review. However, our proposal was exactly to answer these questions directed by the reviewer, carrying out this study design with two proposals of interventions recommended as conservative treatment, in order to, in fact, better recommend the effectiveness of using only spinal brace or its combination with specific exercises, mainly, for Cobb angle reduction and gait and balance data, in the same clinical trial. And our results showed that the spinal brace, in short and long-term, combined with specific exercises in adolescents with idiopathic scoliosis proved to be effective for correcting the scoliotic curvature (cobb angle), improving the anteroposterior and mediolateral body balance, and reducing the plantar load on the rearfoot region during gait, demonstrating effective mechanical action on the spine. To better explain the data, we have added a discussion paragraph to better explain these findings with the literature. Page 14 (line 456-470). Underlined parts. 

Reviewer 3 Report

Respected Authors,

I would like to congratulate you all on your contribution to the scientific community. After reading this manuscript, I felt that this current version must be enhanced, so that the reader shall enjoy it while reading the manuscript.

1.    Abstract represents the manuscript appropriately.

2.    Every sentence is long. Break it into small sentences. The whole conclusion is made up of a single sentence!

3.    Use simple English to frame sentences. I re-read every sentence 2-3 times to understand. Readers might lose interest. Some grammatical errors were also present.

4.    The title can be cut shot into “Effect of short and long term spinal orthopaedic brace in adolescent idiopathic scoliosis”.

5.    Use superscripts and subscripts in appropriate places.

6.    Where are the figures 3, 4, and 5 mentioned in line 215?

7.    Line 233, Insert Figure 1?

8.    Based on what reference short term was considered as 24 hrs? and the long term as 6 months?

9.    Lines 319-320. Is that content correct?

10. Line 335, spinhal

 Regards.

Author Response

São Paulo, 14th of July 2022

Dear Chief Editor

Prof. Dr. Edgaras Stankevičius

Editor and Editorial Board Member of Medicina

We, the authors, would like to resubmit the paper The effect of short and long-term spinal brace with and without exercise on spine, balance and gait in adolescent with idiopathic scoliosis” (1761918-1) in a revised form, as suggested. We are also sending a covering letter responding to the reviewer’s comments on a point-by-point basis. Bellow, we followed your comments and answered each one also in a point-by-point basis. The authors would like to thank the editor for the careful revision and constructive comments/suggestions on our manuscript that certainly contributed for a better version of it.

Yours sincerely,

The authors

Responses to the Reviewer #3:

Submission: MEDICINA (ID 1761918-1)

Reviewer #3:

We thank the reviewer for the constructive comments on our manuscript. Your suggestions and remarks have helped us to reflect on the manuscript and make a better version. We appreciate your suggestions and carefully considered every one of your comments and we made the appropriate changes. Below, we responded to your remarks on a point-by-point basis and inserted the corrections to the new version of the manuscript (underlined parts). Some of your comments will be discussed here. First your comment is given in bold; subsequently we provide our answer.

  1. Respected Authors. I would like to congratulate you all on your contribution to the scientific community. After reading this manuscript, I felt that this current version must be enhanced, so that the reader shall enjoy it while reading the manuscript.

Answer: We appreciate your comment and all your attention and care in reviewing this manuscript. We are very honored by your review.

  1. Abstract represents the manuscript appropriately.

Answer: We strongly appreciate and thank reporting Editor. 

  1. Every sentence is long. Break it into small sentences. The whole conclusion is made up of a single sentence!

Answer: We agreed and appreciate your comment. We rewrite the conclusion with shorter sentences for better understanding. Page 1 (line 35-39); Page 14 (line 485-489). Underlined parts. 

  1. Use simple English to frame sentences. I re-read every sentence 2-3 times to understand. Readers might lose interest. Some grammatical errors were also present.

Answer: We appreciated your comment and strongly apologize for the grammatical and typographical errors in the manuscript. We have submitted the manuscript to a professional service for reviewing English as a second language (Scribendi). We hope it achieves the standards of this respectful Journal.

  1. The title can be cut shot into “Effect of short and long term spinal orthopaedic brace in adolescent idiopathic scoliosis”.

Answer: We strongly appreciate the reviewer comment. However, in addition to spinal brace treatment, exercises were also performed to adjust the spine and also gait and balance. In this rationale, we tried to make the title shorter. We hope we have responded to the reviewer's request. Page 1 (line 2-3). Underlined parts. 

“The effect of short and long-term spinal brace with and without exercise on spine, balance and gait in adolescent with idiopathic scoliosis.”

  1. Use superscripts and subscripts in appropriate places.

Answer: We agreed and appreciate your comment. We have corrected this error in text. Page 5 (line 221 and 223); Page 13 (line 396-400). Underlined parts. 

  1. Where are the figures 3, 4, and 5 mentioned in line 215?

Answer: We appreciate your comment. The figures (specific exercise program) were submitted, we do not understand why it did not appear to the reviewer. We apologize for that. Figures have been added to the text. Page 7 (figure 3); Page 8 (figure 4) and Page 9 (figure 5). Underlined parts. 

  1. Line 233, Insert Figure 1?

Answer: We agreed and appreciate your comment. We have corrected this error in text. Page 4 (line 221 and 223); Page 10 (line 323). Underlined parts. 

  1. Based on what reference short term was considered as 24 hrs? and the long term as 6 months?

Answer: We agreed and appreciate your comment. We followed your request and added the references that supported the short and long-term intervention in the text. Page 6 (line 244-247); Page 7 (line 281). Underlined parts. 

  1. Lines 319-320. Is that content correct?

Answer: We agreed and appreciate your comment. We have corrected this error in text. Page 13 (line 415-416). Underlined parts. 

  1. Line 335, spinhal

Answer: We agreed and appreciate your comment. We have corrected this error in text. Page 13 (line 430). Underlined parts. 

Round 2

Reviewer 2 Report

1. Page 2, Line 56 the word postural has been repeated twice

2. We appreciate the efforts put in by the authors. However the intro is very long (spanning 2 pages). Can they cut down on the initial 3 paragraphs involving general discussion on AIS?

3. Methods section has been well modified and elaborated

4. The other recommendations  have been well-modified. The manuscript reads well now

Author Response

São Paulo, 21th of July 2022

Dear Chief Editor

Prof. Dr. Edgaras Stankevičius

Editor and Editorial Board Member of Medicina

We, the authors, would like to resubmit the paper The effect of short and long-term spinal brace with and without exercise on spine, balance and gait in adolescent with idiopathic scoliosis” (1761918-2) in a revised form, as suggested. We are also sending a covering letter responding to the reviewer’s comments on a point-by-point basis. Bellow, we followed your comments and answered each one also in a point-by-point basis. The authors would like to thank the editor for the careful revision and constructive comments/suggestions on our manuscript that certainly contributed for a better version of it.

Yours sincerely,

The authors

Responses to the Reviewer #2:

Submission: MEDICINA (ID 1761918-1)

 Reviewer #2:

We thank the reviewer for the constructive comments on our manuscript. Your suggestions and remarks have helped us to reflect on the manuscript and make a better version. We appreciate your suggestions and carefully considered every one of your comments and we made the appropriate changes. Below, we responded to your remarks on a point-by-point basis and inserted the corrections to the new version of the manuscript (underlined parts). Some of your comments will be discussed here. First your comment is given in bold; subsequently we provide our answer.

  1. We appreciate the efforts put in by the authors. However, the intro is very long (spanning 2 pages). Can they cut down on the initial 3 paragraphs involving general discussion on AIS?

Answer: We appreciate your suggestion and agree. Thus, we have reduced the first three paragraphs of the introduction to just one paragraph. Now, the introductory session contains 4 paragraphs with the contextualization of the study theme and a paragraph with the justification and objective of the present study. Thank you very much for your suggestion. Page 1 (line 44-46); Page 2 (line 47-58). Underlined parts.

  1. Page 2, Line 56 the word postural has been repeated twice.

Answer: We agree with the reviewer and apologize for this error. We delete the repeated word. Page 2 (line 50). Underlined parts.

  1. Methods section has been well modified and elaborated.

Answer: We really appreciate all your attention, care and contribution to the improvement of our manuscript. Really, your review made a big difference to the good reading of the manuscript. We would like to reiterate our thanks for your review.

  1. The other recommendations have been well-modified. The manuscript reads well now.

Answer: Thank you very much for your contribution. 
